# Associations of parental depression during adolescence with cognitive development in later life in China: A population-based cohort study

Zhihui Li[1,2‡]*, Wenjuan Qin[3‡], Vikram Patel[2,4]*

**1** Vanke School of Public Health, Tsinghua University, Beijing, China, **2** Department of Global Health and Population, Harvard TH Chan School of Public Health, Boston, Massachusetts, United States of America, **3** College of Foreign Languages and Linguistics, Fudan University, Shanghai, China, **4** Department of Global Health and Social Medicine, Harvard Medical School, Boston, Massachusetts, United States of America

‡ These authors share first authorship on this work.

* Zhihuili@mail.tsinghua.edu.cn (ZL); Vikram_Patel@hms.harvard.edu (VP)

## Abstract

### Background

Prior research has underscored negative impacts of perinatal parental depression on off-spring cognitive performance in early childhood. However, little is known about the effects of parental depression during adolescence on offspring cognitive development.

### Methods and findings

This study used longitudinal data from the nationally representative China Family Panel Studies (CFPS). The sample included 2,281 adolescents aged 10–15 years (the median age was 13 years with an interquartile range between 11 and 14 years) in 2012 when their parents were surveyed for depression symptoms with the 20-item Center for Epidemiologic Studies Depression Scale (CES-D). The sample was approximately balanced by sex, with 1,088 females (47.7%). We examined the associations of parental depression in 2012 with offspring cognitive performance (measured by mathematics, vocabulary, immediate word recall, delayed word recall, and number series tests) in subsequent years (i.e., 2014, 2016, and 2018) using linear regression models, adjusting for various offspring (i.e., age, sex, and birth order), parent (i.e., parents' education level, age, whether living with the offspring, and employment status), and household characteristics (i.e., place of residence, household income, and the number of offspring). We found parental depression during adolescence to be significantly associated with worse cognitive performance in subsequent years, in both crude and adjusted models. For example, in the crude models, adolescents whose mothers had depression symptoms in 2012 scored 1.0 point lower (95% confidence interval [CI]: −1.2 to −0.8, $p < 0.001$) in mathematics in 2014 compared to those whose mothers did not have depression symptoms; after covariate adjustment, this difference marginally reduced to 0.8 points (95% CI: −1.0 to −0.5, $p < 0.001$); the associations remained robust after further adjusting for offspring earlier cognitive ability in toddlerhood (−1.2, 95% CI: −1.6, −0.9, $p <$

**Data Availability Statement:** All data files are available from https://opendata.pku.edu.cn/dataset.xhtml?persistentId=doi:10.18170/DVN/45LCSO with application.

**Funding:** WQ received funding from Chinese Ministry of Education Humanities and Social Science Foundation (Grant number: 19YJC740058). The funders had no role in study design, data collection and analysis, decision to publish, or preparation of the manuscript.

**Competing interests:** VP is a member of PLOS Medicine's editorial board.

**Abbreviations:** CES-D, Center for Epidemiologic Studies Depression Scale; CFPS, China Family Panel Studies; CI, confidence interval; DSM-III, Diagnostic and Statistical Manual of Mental Disorders-III; IQ, intelligence quotient; IRB, Institutional Review Board; STROBE, Strengthening the Reporting of Observational Studies in Epidemiology.

0.001), offspring cognitive ability in 2012 (−0.6, 95% CI: −0.8, −0.3, $p < 0.001$), offspring depression status (−0.7, 95% CI: −1.0, −0.5, $p < 0.001$), and parents' cognitive ability (−0.8, 95% CI: −1.2, −0.3, $p < 0.001$). In line with the neuroplasticity theory, we observed stronger associations between maternal depression and mathematical/vocabulary scores among the younger adolescents (i.e., 10–11 years) than the older ones (i.e., 12–15 years). For example, the association between maternal depression and 2014 vocabulary scores was estimated to be −2.1 (95% CI: −2.6, −1.6, $p < 0.001$) in those aged 10–11 years, compared to −1.2 (95% CI: −1.6, −0.8, $p < 0.001$) in those aged 12–15 years with a difference of 0.9 (95% CI: 0.2, 1.6, $p = 0.010$). We also observed a stronger association of greater depression severity with worse mathematical scores. The primary limitations of this study were the relatively high attrition rate and residual confounding.

## Conclusions

In this study, we observed that parental depression during adolescence was associated with adverse offspring cognitive development assessed up to 6 years later. These findings highlight the intergenerational association between depression in parents and cognitive development across the early life course into adolescence.

## Author summary

### Why was this study done?

- Depression is the leading mental health–related contributor to the Global Burden of Disease.

- A large body of research has underscored the negative association between parental depression in the perinatal period and offspring cognitive performance in early childhood.

- Little is known about the association between parental depression and offspring cognitive performance in adolescence, a critically sensitive period for neurodevelopment.

### What did the researchers do and find?

- Using a nationally representative sample from China with 2,281 adolescents aged 10–15 years in 2012, we explored the association between parental depression status and offspring cognitive performance (e.g., math, vocabulary, and working memory) during adolescence over a period of 6 years.

- We found that both maternal depression and paternal depression were associated with worse offspring cognitive performance up to 6 years later and across a range of cognitive performance tests, consistent with the theory emphasizing the importance of both parents in shaping cognitive outcomes during adolescence.

- The associations remained robust even after adjusting for a wide range of confounding variables (e.g., sex, parents' educational level, etc.), offspring historic cognitive

performance in toddlerhood, and in 2012, offspring depression status and parents' cognitive ability.

- Stronger associations were found between maternal depression and younger adolescents' cognitive performance (i.e., 10–11 years) than older ones (i.e., 12–15 years), which was aligned with neuroplasticity being more marked in younger adolescents.

### What do these findings mean?

- Our results are consistent with the inter-generational association between parental depression and cognitive development established in early childhood period, but extend such association well into adolescence when the associations are visible for both parents.
- This evidence furthers the understanding of the life course determinants of cognitive development and emphasizes the importance of preventing and treating parental depression beyond early childhood into adolescence.

## Introduction

Depression is one of the leading causes of disability worldwide, with adverse consequences not only on individuals but also on their family members, particularly offspring. Maternal depression during the perinatal period has constantly been found to hinder early childhood cognitive development [1,2]. Paternal depression, though relatively less studied, demonstrates similar adverse effects [3]. Recent studies also document that the impact of parental depression during the perinatal period could persist throughout childhood and adolescence. For instance, these studies show that perinatal maternal depression is significantly associated with lower adolescents' intelligence quotient (IQ) [4] and poorer academic achievement [5,6].

Emerging evidence shows that adolescence is a sensitive period of brain development with heightened neuroplasticity [7,8]. As the brain undergoes substantial changes in adolescence, it is considered a critical and sensitive period of cognitive development, which is also vulnerable to environmental influences, such as stressful life events or cognitive training [7,9]. Despite its established role in early childhood, little is known about the effects of parental depression on offspring cognitive development during adolescence. The only published study we could identify so far shows that parental depression diagnosed when children were aged 11 to 16 years is negatively associated with school performance at age 16 in Sweden [10].

Another limitation of the existing evidence base is that the vast majority of studies regarding parental depression have focused on mothers; while it may be argued that this is an appropriate focus for early child development given the unique role the mother plays in nurturing this phase of life, both parents play equally important roles in adolescence. Indeed, there is increasing evidence indicating that paternal depression independently affects parenting behaviors and children's development outcomes [11,12]. However, little is known about the association of paternal depression with offspring cognitive development.

This study aims to investigate the association of both maternal and paternal depression with offspring cognitive development during adolescence. We also aim to explore the duration of such associations (i.e., how far into the developmental life course these could be observed), their association with timing (i.e., whether the associations are stronger if the adolescent

offspring is younger, in line with the current understanding of the critical sensitive periods of neuroplasticity), and their association with dosage (i.e., whether the associations are stronger for more severe depressive syndromes).

## Methods

For the current study, the analysis plan was drafted in January 2020. During the revision process, in response to editors' and reviewers' comments, 4 sets of sensitivity analyses were added to address potential confounding issues and attribution bias (S1 Text). This study was reported as per the Strengthening the Reporting of Observational Studies in Epidemiology (STROBE) guideline (S1 STROBE Checklist).

### Data source and study sample

Data used in this study were from the China Family Panel Studies (CFPS). CFPS is a nationally representative longitudinal survey conducted by the Institute of Social Science Survey at Peking University in collaboration with the Survey Research Center at the University of Michigan [13]. CFPS was first launched in 2010 with subsequent rounds of data collected in 2012, 2014, 2016, and 2018. The target sample of CFPS consists of around 16,000 Chinese households, representing 95% of the Chinese population [13]. CFPS collected information at the individual, household, and community levels on a range of topics, including parents' physical and mental health, education and work status, economic activities, and offspring cognitive performance. Using a multistage probability-proportional-to-size sampling process, CFPS selected 144 counties from 25 provinces and 32 townships in Shanghai as the primary sampling unit. A total of 640 communities were randomly sampled within counties, and then 25 households were randomly selected from each community. All eligible households and household members were participants of the survey [13].

Our study used the 2012 survey as the baseline when depression measurement for at least 1 parent was available. Of all the households included in the CFPS data, there were 2,406 offspring aged 10 to 15 years in the CFPS 2012 survey. Among them, we excluded 135 offspring with both parents' depression measures marked as "unknown," "refuse to answer," or "inapplicable". Thus, the final sample comprised 2,281 offspring.

To assess the impact of parental depression on offspring cognitive development, we set up the data in a panel form with parents' depression status measured in 2012 and offspring cognitive performance measured in 2014, 2016, and 2018. Among the 2,281 offspring identified in 2012, we successfully followed up 85.9% of the offspring in 2014, 72.1% in 2016, and 63.7% in 2018 (Table 1). The number of offspring with valid cognitive measures also differed by survey waves. For example, among the 2,281 offspring, 320 of them were lost of follow-up in 2014, and another 250 of them had their vocabulary test score marked as "unknown" or "inapplicable". As a result, in total, we included 1,711 offspring for the analysis on the vocabulary test in 2014. See Fig 1 for details.

This project used publicly accessible secondary data obtained from the CFPS website (https://www.isss.pku.edu.cn/cfps/en/data/public/index.htm) which excluded all identifiable information about individual participants. According to the Institutional Review Board (IRB) guideline, the research activities did not meet the regulatory definition of human subjects research. As such, IRB review was not required. The Harvard Longwood Campus IRB allows researchers to self-determine when their research does not meet the requirements for IRB oversight via guidance online regarding when an IRB application is required and the IRB Decision Tool [14]. This study did not require additional specific approval from an ethics committee.

**Table 1. Attrition status and summary statistics for key characteristics by year.**

| | 2012 | 2014 | 2016 | 2018 |
|---|---|---|---|---|
| **No. of offspring aged 10–15 in 2012** | **2,281** | **1,960** | **1,644** | **1,452** |
| **Offspring characteristics** | | | | |
| Age in 2012[#] | 12.60 (0.04) | 12.53 (0.04) | 12.48 (0.04) | 12.46 (0.04) |
| Sex (Female) | 1,088 (47.7%) | 941 (48.0%) | 776 (47.2%) | 673 (46.4%) |
| Birth order | | | | |
| First | 1,626 (71.3%) | 1,400 (71.4%) | 1,181 (71.8%) | 1,041 (71.7%) |
| Second | 545 (23.9%) | 465 (23.7%) | 390 (23.7%) | 349 (24.0%) |
| Third or more | 110 (4.8%) | 95 (4.9%) | 73 (4.4%) | 62 (4.3%) |
| **Parents' characteristics** | | | | |
| Maternal education level in 2012 | | | | |
| ≤6 years | 1,194 (52.4%) | 1,033 (52.7%) | 864 (52.6%) | 767 (52.8%) |
| 7–9 years | 540 (23.7%) | 490 (25.0%) | 417 (25.4%) | 379 (26.1%)* |
| ≥10 years | 547 (24.0%) | 437 (22.3%)* | 363 (22.1%)* | 306 (21.1%)** |
| Paternal education level in 2012 | | | | |
| ≤6 years | 826 (36.2%) | 702 (35.8%) | 590 (35.9%) | 525 (36.2%) |
| 7–9 years | 639 (28.0%) | 565 (28.8%) | 473 (28.8%) | 425 (29.3%) |
| ≥10 years | 816 (35.8%) | 693 (35.4%) | 581 (35.3%) | 502 (34.6%) |
| Maternal age in 2012 | | | | |
| ≤35 | 523 (22.9%) | 447 (22.8%) | 378 (23.0%) | 347 (23.9%) |
| 36–40 | 940 (41.2%) | 801 (40.9%) | 683 (41.6%) | 591 (40.7%) |
| 41–45 | 593 (26.0%) | 522 (26.6%) | 429 (26.1%) | 385 (26.5%) |
| ≥46 | 225 (9.9%) | 190 (9.7%) | 154 (9.4%) | 129 (8.9%) |
| Paternal age in 2012 | | | | |
| ≤35 | 286 (12.5%) | 254 (13.0%) | 212 (12.9%) | 203 (14.0%) |
| 36–40 | 915 (40.1%) | 773 (39.4%) | 666 (40.5%) | 583 (40.2%) |
| 41–45 | 690 (30.3%) | 598 (30.5%) | 500 (30.4%) | 433 (29.8%) |
| ≥46 | 390 (17.1%) | 335 (17.1%) | 266 (16.2%) | 233 (16.1%) |
| Offspring lived together with the mother for more than 8 months in the past 12 months, 2012 | 1,618 (70.9%) | 1,411 (72.0%) | 1,193 (72.6%) | 1,070 (73.7%)* |
| Offspring lived together with the father for more than 8 months in the past 12 months, 2012 | 1,378 (60.4%) | 1,201 (61.3%) | 1,016 (61.8%) | 922 (63.5%)** |
| Mother was employed or self-employed in 2012 | 1,893 (83.0%) | 1,658 (84.6%) | 1,395 (84.9%) | 1,227 (84.5%) |
| Father was employed or self-employed in 2012 | 2,018 (88.5%) | 1,747 (89.1%) | 1,473 (89.6%) | 1,315 (90.6%) |
| **Household characteristics** | | | | |
| Lived in rural areas in 2012 | 1,717 (75.3%) | 1,483 (75.7%) | 1,237 (75.2%) | 1,107 (76.2%) |
| Household income per capita in 2012 (¥ USD), log scale[#] | 6.80 (0.03) | 6.80 (0.03) | 6.80 (0.03) | 6.77 (0.03) |
| Number of children in the household in 2012[#] | 2.08 (0.02) | 2.08 (0.02) | 2.07 (0.02) | 2.06 (0.02) |
| **Offspring earlier cognitive ability for sensitivity analysis[φ]** | | | | |
| Age to speak a complete sentence in month ($N = 1,426$)[#] | 24.68 (0.19) | 24.53 (0.20) | 24.51 (0.22) | 24.67 (0.23) |
| Age to count from 1 to 10 in month ($N = 1,386$)[#] | 35.19 (0.25) | 35.36 (0.25) | 35.18 (0.28) | 35.26 (0.29) |

**Note**: The asterisks represent the significance in the comparison between the values in that year and the value in 2012.

*$p < 0.05$

**$p < 0.01$, ***$p < 0.001$.

[#]We placed standard errors in the parenthesis following the mean values for these items.

[φ]Data on child earlier cognitive ability were from the survey conducted in 2010.

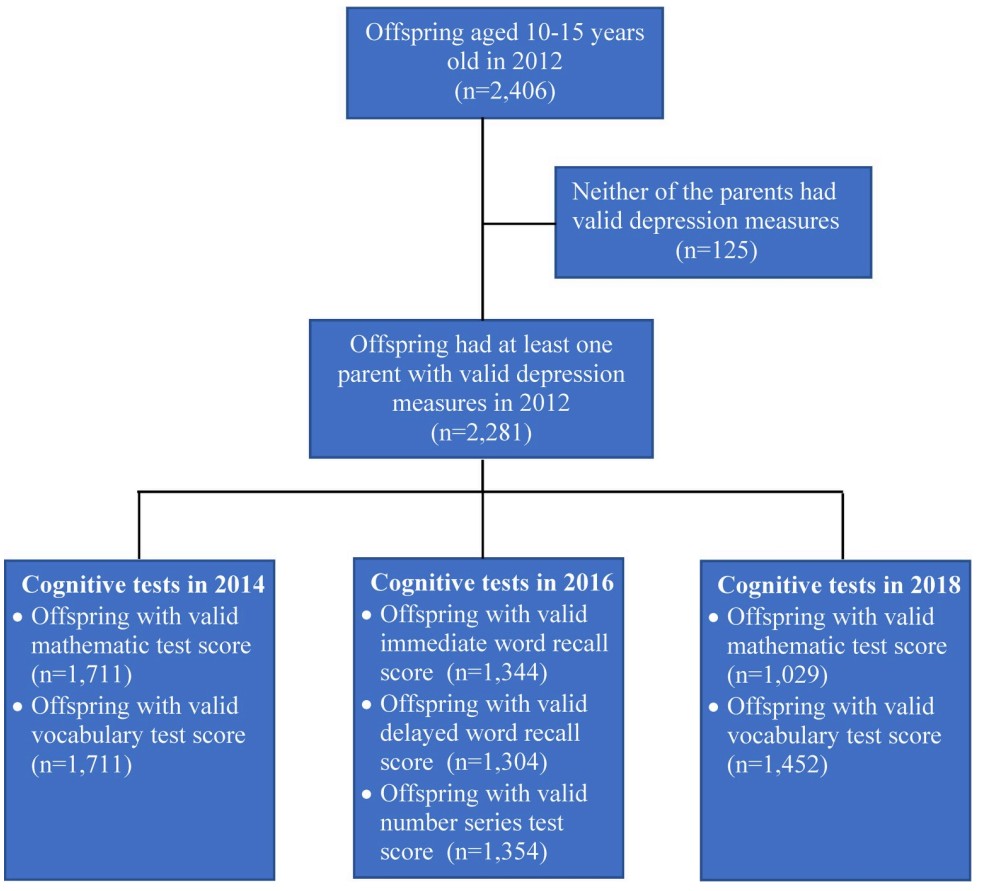

**Fig 1. Flowchart of sample selection.**

## Exposure measure

Parental depression status was measured using a 20-item full version of the Center for Epidemiologic Studies Depression Scale (CES-D) in the 2012 wave. The CES-D has been widely used in various global contexts [15,16] to screen for depression symptoms and has been previously validated for use in the Chinese population [17,18]. In CFPS 2012, parents rated the 20-symptom items over the past week (e.g., I feel lonely) on a 4-point scale of 0 to 3, which were "almost never (less than 1 day)," "sometimes or occasionally (1 or 2 days a week)," "frequently (3 or 4 days a week)," and "most of the time (5 days or more a week)." Scores for each parent ranged from 0 (lowest) to 60 (highest). Following the traditional standard proposed by the scale developer [19] and evidence from validation studies [20–22], we defined parents who scored 16 or higher on CES-D as the ones with depression symptoms; we further categorized those with scores of 16 to 24 to be with "moderate" depression symtoms and those with scores of 25 or higher to be with "severe" depression symtoms (S1 Table). These cutoff scores have been validated with Diagnostic and Statistical Manual of Mental Disorders-III (DSM-III) criteria for clinical depression and has been used in diverse populations [20–22].

## Outcome measures

The CFPS has developed 2 sets of tests to measure the cognitive performance of individuals aged 10 years or older. Cognitive performance was defined as the composition of "crystalized

intelligence" and "fluid intelligence," which were assessed using different measures in different years [23,24]. Specifically, in the waves of 2010, 2014, and 2018, mathematics and vocabulary tests were conducted to assess "crystallized intelligence," referred to as the ability to use knowledge acquired through past learning, experience, and education [23]. Test items were drawn from the standard curriculums in Chinese primary and secondary schools; in the waves of 2012 and 2016, immediate word recall, delayed word recall, and number series tests were collected to measure "fluid intelligence," referred to as the ability to reason and solve problems in unique and novel situations [23,24]. Both sets of tests have been widely adopted and validated in previous studies to measure cognitive ability among adolescents and adults in the Chinese population [25,26]. The score ranged from 0 to 24 for mathematics test, 0 to 34 for vocabularty test, 0 to 10 for immediate/delayed word recall test, and 0 to 15 for number series test. For all tests, the lowest score was 0; larger number indicated better performance. Test design of each test is listed in S2 Table.

## Covariates

Following previous practices [10,27], we adopted covariates at offspring, parents, and household levels (see Table 1). Offspring characteristics included age, sex, and birth order. Parents' characteristics included parents' education levels, age, whether living together with the offspring for more than 8 months in the past 12 months, and employment status. Except for parents' education level and ages, all other parents' characteristics were dichotomous variables. Household characteristics included place of residence (i.e., urban or rural), household income in log scale, and the number of offspring in the household. In the sensitivity analysis, we further included 2 retrospective cognitive measures in offspring toddlerhood to account for the variability in offspring cognitive ability in earlier stages of life. The 2 covariates included are (1) the age that the offspring began speaking in whole sentences (e.g., "I want to eat"); and (2) the age that the offspring began counting from 1 to 10. Speaking capacities and counting skills are widely used to assess the cognitive performance of young children [28,29]. Yet, since nearly 40% of the sample were missing information on these covariates (38.8% [885/2,281] for age to speak in whole sentence and 39.2% [895/2,281] for age to count from 1 to 10), we only controlled for them in sensitivity analyses.

## Statistical analysis

We first examined descriptive statistics and conducted bivariate comparisons of the distributions of these variables using *t* test by parental depression status. Then, we explored the association between parental depression status and offspring cognitive development with a series of linear regression models: In the crude model, we only controlled for the offspring age, which was time varying. In the fully adjusted model, we further controlled for time-invariant offspring characteristics (i.e., sex and birth order), parents' characteristics (i.e., education levels, age, whether living with the offspring, and employment status), and household characteristics (i.e., place of residence, household income in log scale, and the number of offspring in the household).

Besides the crude and fully adjusted models, we also fit 2 more partially adjusted models to test the robustness of our findings. First, we included all offspring characteristics in the model (but not characteristics of parents or household); second, we included all offspring and parents' characteristics (but not household characteristics).

Additionally, we conducted 3 sets of effect modification analyses: First, we explored the effect modification by offspring age at exposure using fully adjusted linear regression models with the interaction term between offspring age and maternal/paternal depression status.

Then, we used the fully adjusted model with an interaction term between offspring sex and maternal/paternal depression status to examine the effect modification by offspring sex. Last, we stratified the sample by the severity of parental depression and constructed linear regression models (controlling for all offspring, parents, and household characteristics) to examine the association between severity of depression and offspring cognitive development.

We also conducted 6 sets of sensitivity analyses to assess the robustness of our results. We added, respectively, (1) offspring retrospective cognitive measures during their toddlerhood and (2) their cognitive test scores measured in 2012 into the fully adjusted models as proxies of their earlier cognitive ability. (3) We treated parental CES-D score as a continuous variable in both crude and fully adjusted models to test the potential problem for treating the CES-D score as dichotomous. (4) We adopted multiple imputation to impute for observations loss of follow-up or for observations with key covariates missing in 2014, 2016, and 2018. The multiple imputation was carried out using the MI command in Stata (version 14.2) developed by StataCorp, College Station, Texas, United States of America. In our imputation, we included all the variables on offspring characteristics, parents' characteristics, and household characteristics, as well as parental CES-D scores. (5) We added offspring CES-D score measured in 2012 as a covariate to the fully adjusted models to adjust for the generic effect of parental depression. (6) We added parental cognitive scores (i.e., math test and word test scores) measured in 2010 to the fully adjusted models to adjust for the effects of transgenerational transmission of cognitive ability.

All models used robust standard errors clustered at the province level to account for correlation in the exposure and outcomes across provinces. We used Stata (version 14.2) for all analytic procedures. All statistical tests were 2-sided, and $p < 0.05$ was used as the criteria to determine statistical significance.

## Results

Table 1 shows attrition status and summary statistics for key characteristics of the study sample ($n = 2,281$). Among the 2,281 offspring identified in the 2012 survey, 1,960 (85.9%), 1,644 (72.1%), and 1,452 (63.7%) were followed-up in 2014, 2016, and 2018, respectively. There was no significant difference for most baseline parental characteristics between those offspring who were followed up and those who were lost, with the exception of maternal education and whether either parent lived with the offspring for more than 8 months in the past 12 months. We compared the characteristics at baseline by parental depression status in S3 Table. We found that depressed parents (1 or both parents with depression symptoms) had significantly lower education levels, lower household income, and were more likely to live in rural areas and have more offspring. Moreover, offspring in the depression group were more likely to be later-born and were later to begin speaking a complete sentence and counting from 1 to 10 in toddlerhood.

### Parental depression and offspring cognitive development

Both crude and adjusted models demonstrate that parental depression was associated with lower scores in offspring cognitive performance (Tables 2 and 3). Maternal depression was significantly associated with all 7 outcomes in the crude models; the coefficients attenuated marginally after covariate adjustment for most outcomes, but remained statistically significant for all 7 outcomes. For example, in the crude analysis, maternal depression was associated with 1.0 point lower (95% confidence interval [CI]: −1.2, −0.8, $p < 0.001$) mathematics scores in 2014; after covariate adjustment (fully adjusted models), the magnitude of this association attenuated marginally to −0.8 (95% CI: −1.0, −0.5, $p < 0.001$). Yet, there were some exceptions. For

**Table 2. The association of parental depression in 2012 with offspring mathematics and vocabulary test scores in 2014 and 2018[1].**

|  | Coef (95% CI) | *p*-value | Coef (95% CI) | *p*-value | Coef (95% CI) | *p*-value | Coef (95% CI) | *p*-value |
|---|---|---|---|---|---|---|---|---|
| *Crude model* | | | | | | | | |
| Maternal depression in 2012 | −1.002 (−1.233, −0.770) | <0.001 | −0.801 (−1.088, −0.514) | <0.001 | −1.492 (−1.811, −1.173) | <0.001 | −3.471 (−4.152, −2.791) | <0.001 |
| Paternal depression in 2012 | −1.541 (−1.821, −1.261) | <0.001 | −1.388 (−1.749, −1.027) | <0.001 | −0.900 (−1.292, −0.508) | <0.001 | −4.569 (−5.429, −3.709) | <0.001 |
| *Fully adjusted model* | | | | | | | | |
| Maternal depression in 2012 | −0.778 (−1.022, −0.533) | <0.001 | −0.664 (−1.144, −0.184) | 0.009 | −1.511 (−1.849, −1.174) | <0.001 | −2.683 (−3.759, −1.607) | <0.001 |
| Paternal depression in 2012 | −1.356 (−1.652, −1.059) | <0.001 | −1.674 (−2.077, −1.271) | <0.001 | −1.029 (−1.440, −0.617) | <0.001 | −4.006 (−5.247, −2.765) | <0.001 |

Note

1 In the fully adjusted models, we controlled for offspring (i.e., age, sex, and birth order), parents (i.e., maternal and paternal education levels, mother's age and father's age, whether the offspring lived together with the mother, whether the offspring lived together with the father, father's employment status, and mother's employment status) and household (i.e., place of residence, household income in log scale, and number of offspring in the household) characteristics.

CI, confidence interval.

example, in the crude analysis, maternal depression was associated with 0.1 point lower (95% CI: −0.2, −0.0, *p* = 0.025) delayed word recall scores in 2016; the association increased to 0.2 points lower (95% CI: −0.4, −0.1, *p* < 0.001) in the fully adjusted model. We observed similar associations across all 7 outcomes for paternal depression in the fully adjusted models. The partially adjusted models showed consistent results in S4 Table.

## Effect modification

**By age.** We present the effect modification by age in Figs 2–4. Of the 14 associations we analyzed (7 outcomes for paternal and maternal depression, respectively), 6 showed significantly worse cognitive performance among younger adolescents aged 10 to 11 years than the older adolescents aged 12 to 15 years. Four of these associations were found in maternal depression, and 2 were found in paternal depression. For example, the association between maternal depression and mathematical scores in 2014 was estimated to be −1.1 (95% CI: −1.5, −0.7, *p* < 0.001) in those aged 10 to 11 years, comparing to −0.5 (95% CI: −0.8, −0.2, *p* = 0.001)

**Table 3. The association of parental depression in 2012 with offspring immediate word recall, delayed word recall, and number series test scores in 2016[1].**

|  | Immediate word recall (*N* = 1,344) | | Delayed word recall (*N* = 1,304) | | Number series test (*N* = 1,354) | |
|---|---|---|---|---|---|---|
|  | Coef (95% CI) | *p*-value | Coef (95% CI) | *p*-value | Coef (95% CI) | *p*-value |
| *Crude model* | | | | | | |
| Maternal depression in 2012 | −0.121 (−0.201, −0.041) | 0.003 | −0.129 (−0.244, −0.013) | 0.029 | −0.609 (−0.812, −0.406) | <0.001 |
| Paternal depression in 2012 | −0.332 (−0.427, −0.238) | <0.001 | −0.127 (−0.264, 0.010) | 0.069 | −1.559 (−1.801, −1.317) | <0.001 |
| *Fully adjusted model* | | | | | | |
| Maternal depression in 2012 | −0.105 (−0.195, −0.015) | 0.019 | −0.234 (−0.364, −0.103) | <0.001 | −0.581 (−0.802, −0.360) | <0.001 |
| Paternal depression in 2012 | −0.316 (−0.424, −0.207) | <0.001 | −0.183 (−0.338, −0.027) | 0.029 | −1.569 (−1.837, −1.301) | <0.001 |

Note

1 In the fully adjusted models, we controlled for offspring (i.e., age, sex, and birth order), parents (i.e., maternal and paternal education levels, mother's age and father's age, whether the offspring lived together with the mother, whether the offspring lived together with the father, father's employment status, and mother's employment status) and household (i.e., place of residence, household income in log scale, and number of offspring in the household) characteristics.

CI, confidence interval.

 

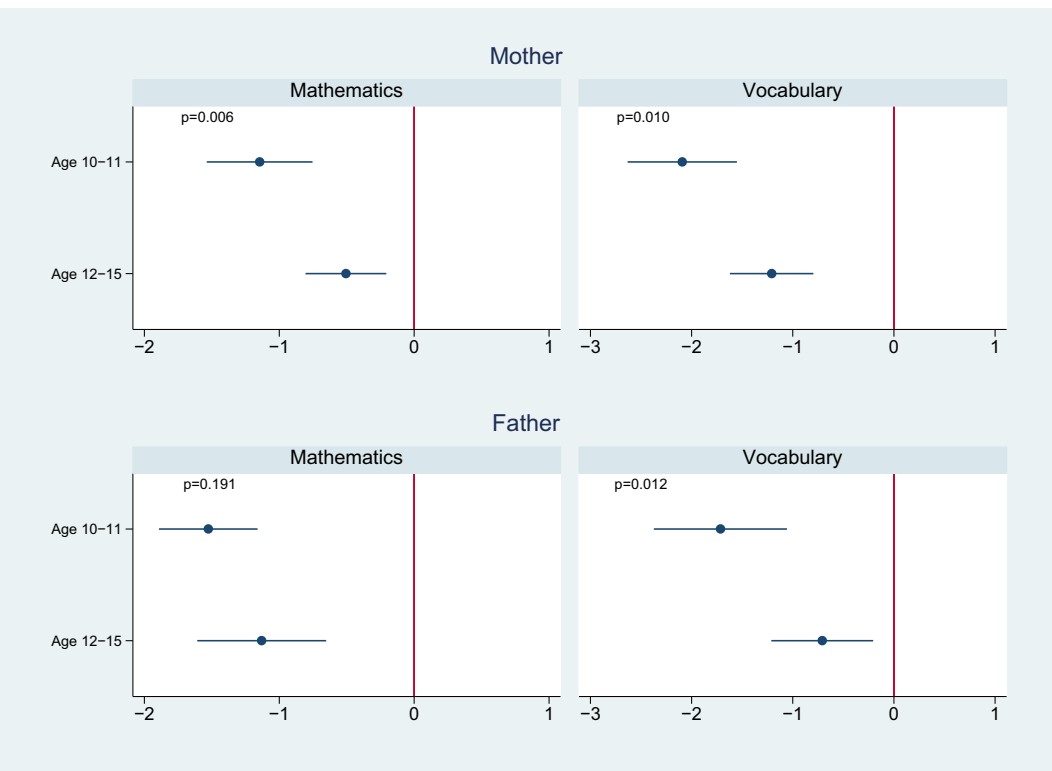

**Fig 2. Effect modification by offspring age in 2012 on the association of parental depression in 2012 with offspring cognitive test scores in 2014. [1] Note:** 1 We controlled for offspring (i.e., age, sex, and birth order), parents (i.e., maternal and paternal education levels, mother's age and father's age, whether the child lived together with the mother, whether the child lived together with the father, father's employment status, and mother's employment status), and household (i.e., place of residence, household income in log scale, and number of children in the household) characteristics.

in those aged 12 to 15 years; the difference between the 2 age groups was 0.6 (95% CI: 0.2, 1.1, $p = 0.006$); the association between maternal depression and vocabulary scores in 2014 was −2.1 (95% CI: −2.6, −1.6, $p < 0.001$) in those aged 10 to 11 years, significantly lower than those aged 12 to 15 years (−1.2, 95% CI: −1.6, −0.8, $p < 0.001$) with a difference of 0.9 (95% CI: 0.2, 1.6, $p = 0.010$); similarly, the association between paternal depression and 2014 vocabulary scores was −1.7 (95% CI: −2.4, −1.0, $p < 0.001$) in those aged 10 to 11 years, compared to −0.7 (95% CI: −1.1, −0.2, $p = 0.006$) in those aged 12 to 15 years; the difference was 1.0 (95% CI: 0.2, 1.8, $p = 0.012$).

**By sex.** We show the effect modification by sex in S5 Table. There was no evidence of differential associations of maternal depression on offspring cognitive performance by sex. However, the associations of paternal depression with some cognitive outcomes appeared different in boys and girls, such that paternal depression had a stronger association with lower 2014 mathematic scores and lower immediate word recall scores in girls than in boys. For example, paternal depression was associated with 0.6 points lower (95% CI: −0.7, −0.4, $p < 0.001$) immediate word recall scores in 2016 in girls, but not in boys (95% CI: −0.2, 0.1, $p = 0.423$); the difference was estimated to be 0.6 points (95% CI: 0.3, 0.9, $p < 0.001$).

**Severity of depression.** We observed significant associations between the hypothesized dose–response effect of maternal depression and 3 cognitive outcomes, including delayed word recall, number sequence, and mathematics scores (Figs 5–7). For example, the magnitudes of association between severe maternal depression and offspring number sequence

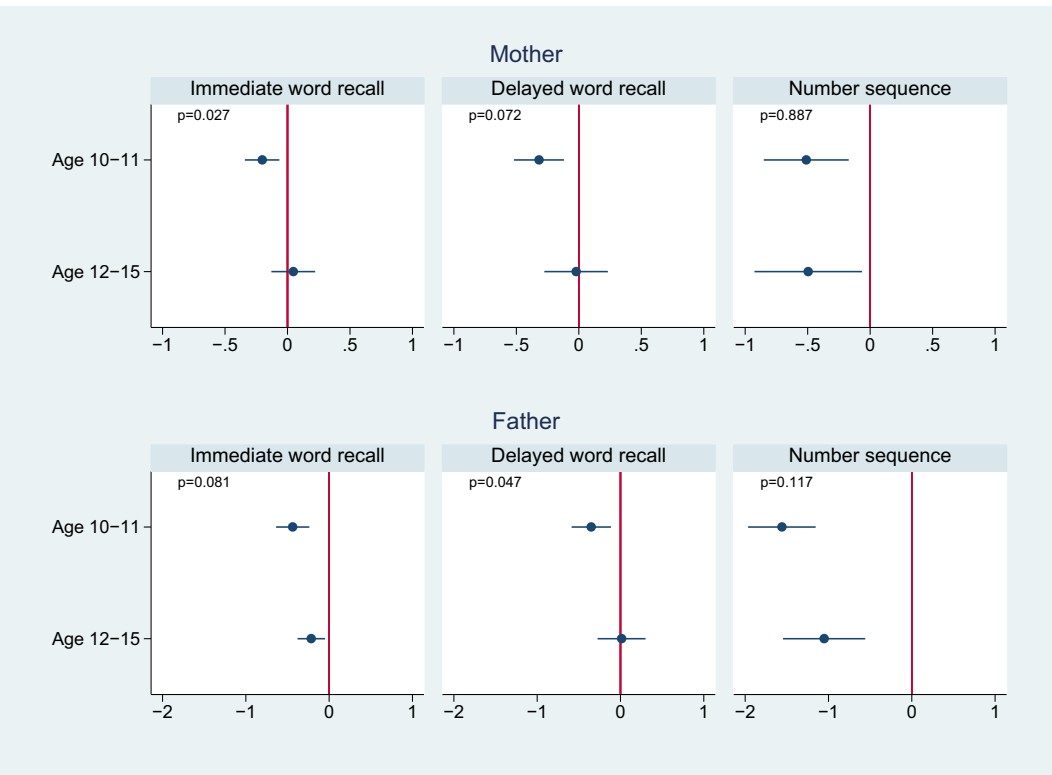

**Fig 3. Effect modification by offspring age in 2012 on the association of parental depression in 2012 with offspring cognitive test scores in 2016. [1] Note:** 1 We controlled for offspring (i.e., age, sex, and birth order), parents (i.e., maternal and paternal education levels, mother's age and father's age, whether the child lived together with the mother, whether the child lived together with the father, father's employment status, and mother's employment status), and household (i.e., place of residence, household income in log scale, and number of children in the household) characteristics.

scores in 2016 (−2.7, 95% CI: −4.1, −1.3, $p < 0.001$) were significantly larger than that for moderate maternal depression (−0.3, 95% CI: −1.2, 0.6, $p = 0.569$); the difference between the 2 groups were estimated to be 2.4 (95% CI: 1.0, 3.9, $p = 0.001$). The effect sizes for paternal depression were smaller, and none reached statistical significance.

## Sensitivity analyses

Results from the sensitivity analyses are shown in Table 4 and S6–S9 Tables. In the first set of sensitivity analysis, we added the variables measuring offspring's retrospective cognitive ability during toddlerhood into the fully adjusted models, and majority of the results remained consistent as above. For example, the association between maternal depression and 2014 mathematics scores in this model was −1.2 (95% CI: −1.6, −0.9, $p < 0.001$), similarly as the result esimtaed in the fully adjusted model above (−0.8, 95% CI: −1.0, −0.5, $p < 0.001$). The only exception was that the association between maternal depression and delayed word recall in 2016 weakened and became nonsignificant in this model (−0.2, 95% CI: −0.3, 0.0, $p = 0.094$) (Table 4).

In the second sensitivity analysis, we added the variables measuring offspring's cognitive ability measured in 2012 into the fully adjusted models. Although most coefficients attenuated comparing to those in the fully adjusted models, they all remained statistically significant. For example, the association beween maternal depression and 2014 mathematics scores in this model was estimated to be −0.6 (95% CI: −0.8, −0.3, $p < 0.001$) (Table 4).

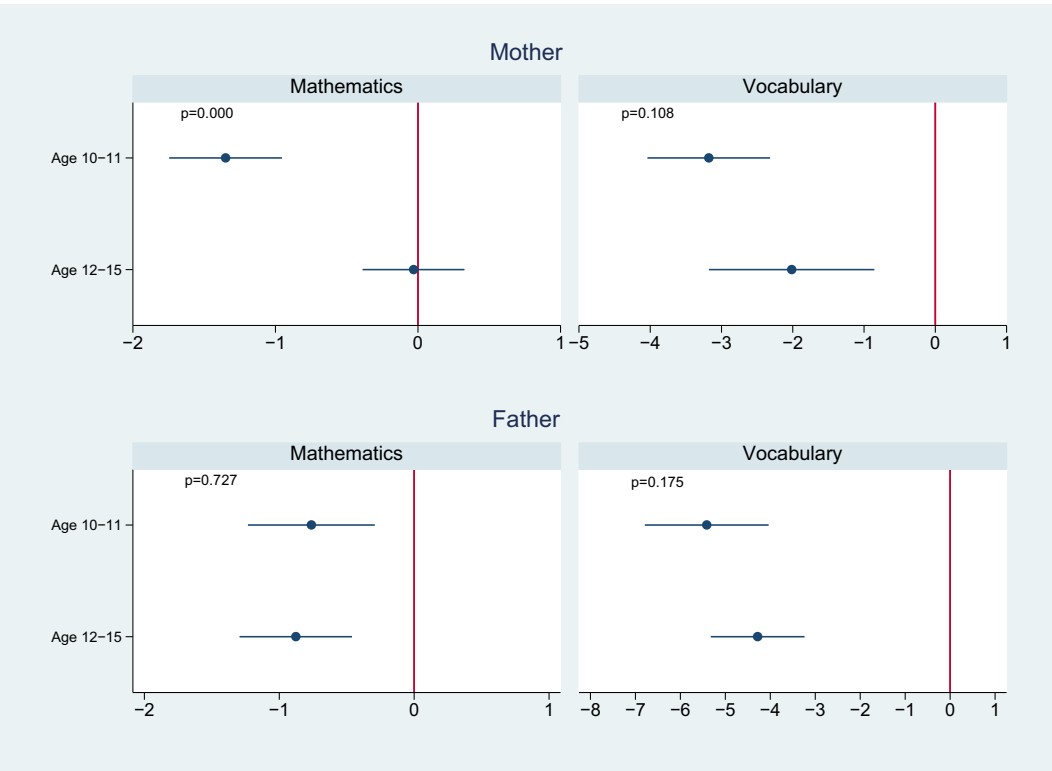

**Fig 4. Effect modification by offspring age in 2012 on the association of parental depression in 2012 with offspring cognitive test scores in 2018.** [1] **Note**: 1 We controlled for offspring (i.e., age, sex, and birth order), parents (i.e., maternal and paternal education levels, mother's age and father's age, whether the child lived together with the mother, whether the child lived together with the father, father's employment status, and mother's employment status), and household (i.e., place of residence, household income in log scale, and number of children in the household) characteristics.

In the third sensitivity analysis, we treated parental CES-D score as a continuous variable and investigated the effect size as the CES-D score increased by 1 unit. The results were consistent with the prior analyses when we treated CES-D score as dichotomous. For example, in the fully adjusted model, we found that an increment of 1 unit CES-D score in mothers was associated with a decrement of 0.09 points (95% CI: −0.11, −0.07, $p < 0.001$) mathematics scores in 2014 (S6 Table).

In the fourth sensitivity analysis, we imputed the test scores and other key covariates for individuals loss of follow-up over years or with no valid data. The results remained consistent (S7 Table). In the fifth set of sensitivity analysis, we added offspring CES-D score measured in 2012 to the fully adjusted models. The results still remained robust for most outcomes. For example, after adding offspring CES-D into the fully adjusted model, the association beween maternal depression and 2014 mathematics scores was −0.7 (95% CI: −1.0, −0.5, $p < 0.001$) (S8 Table). In the sixth set of sensitivity analysis, we added parental cognitive scores measured in 2010, respectively, to the fully adjusted models. The results remained consistent for most outcomes. For example, the association beween maternal depression and 2014 mathematics scores was 0.8 (95% CI: −1.2, −0.3, $p < 0.001$) in this model. One exception was that after adjusting for 2010 parental cognitive scores, the association between maternal depression and number series test was no longer significant (S9 Table).

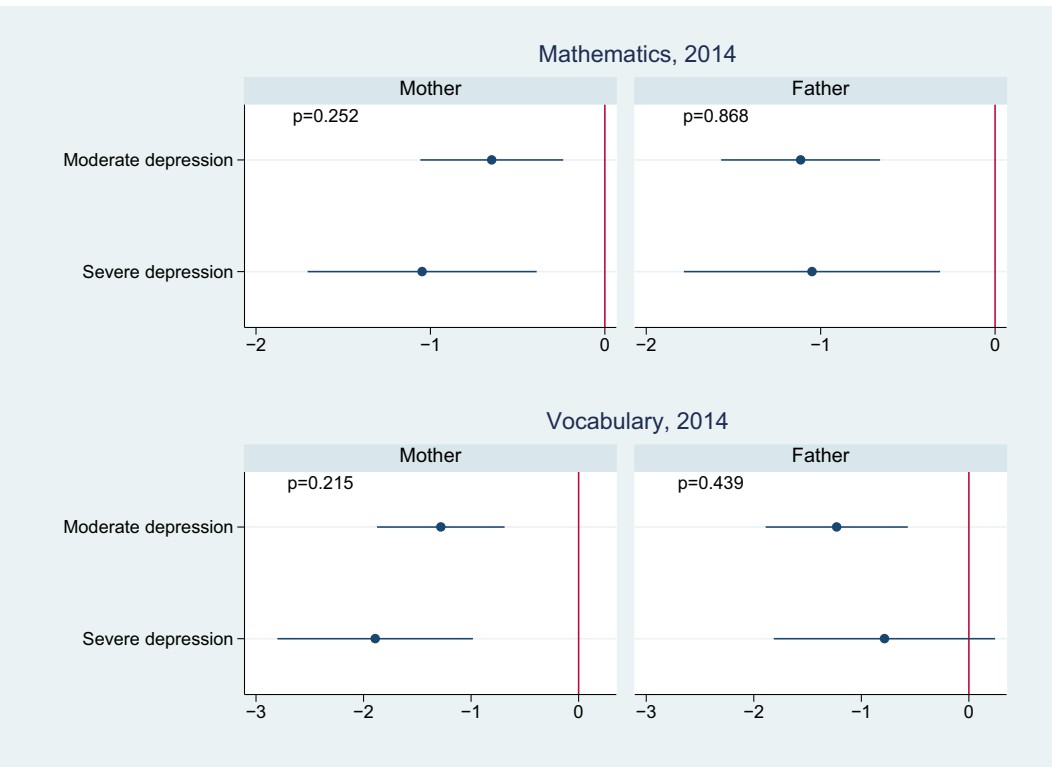

**Fig 5. Effect modification by severity of parental depression in 2012 on the association of parental depression in 2012 with offspring cognitive test scores in 2014.** [1] **Note:** 1 We controlled for offspring (i.e., age, sex, and birth order), parents (i.e., maternal and paternal education levels, mother's age and father's age, whether the child lived together with the mother, whether the child lived together with the father, father's employment status, and mother's employment status), and household (i.e., place of residence, household income in log scale, and number of children in the household) characteristics.

## Discussion

Using a population-based cohort in China, we found that both maternal depression and paternal depression diagnosed when offspring were 10 to 15 years were associated with worse cognitive performance up to 6 years later. Even after adjusting for a wide range of concurrent and historic confounding covariates, these associations were consistent for most cognitive outcomes. Moreover, we observed evidence of effect modification by the age of exposure, suggesting the association between parental depression and offspring cognitive performance to be stronger if the exposure occurred during the earlier years of adolescence (i.e., 10 to 11 years). Offspring sex did not modify the associations of maternal depression, and results for paternal depression were inconsistent. Finally, we found some evidence of a dose–response relationship for maternal depression with greater severity associated with worse cognitive outcomes.

Most research studying the effects of parental depression on offspring cognitive outcomes have been limited to the early childhood period, with a particular focus showing how maternal depression can negatively affect early cognitive development. However, little is known about the effects of parental depression later in the life course, particularly during offspring adolescence, a critical period for cognitive development. Our study addresses this knowledge gap using a nationally representative sample from China. We not only confirm the strong and consistent associations between both maternal and paternal depression and offspring cognitive development during adolescence, but also contribute to the growing evidence suggesting that these effects may be causal [5,30]:

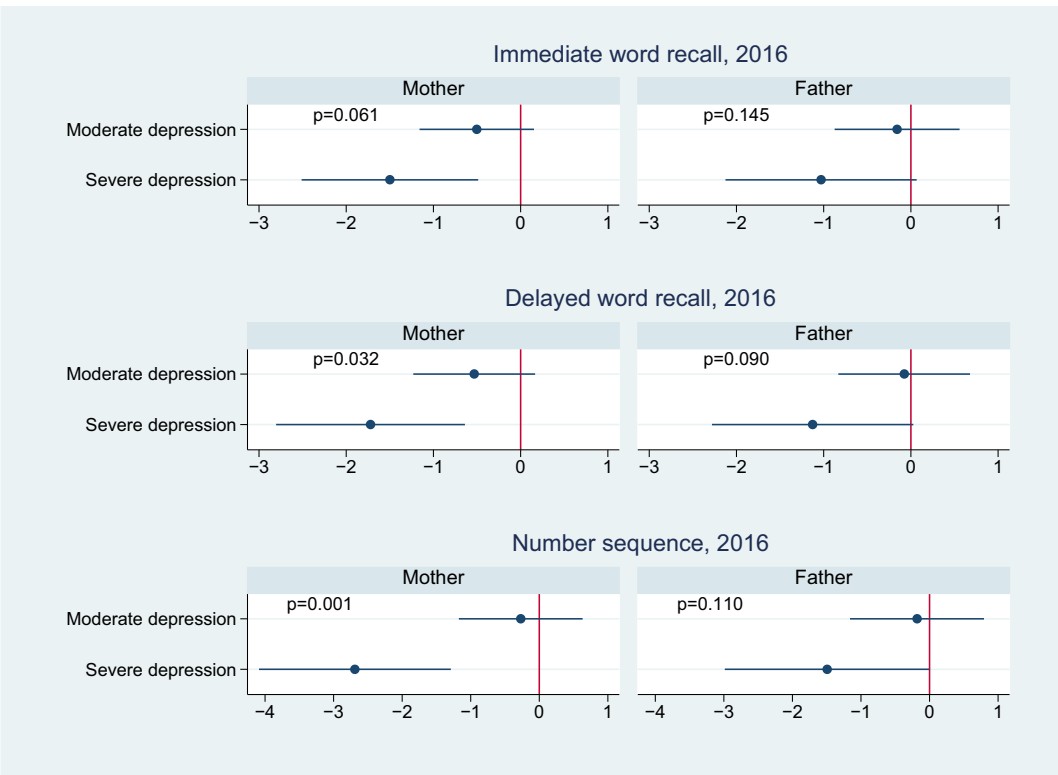

**Fig 6. Effect modification by severity of parental depression in 2012 on the association of parental depression in 2012 with offspring cognitive test scores in 2016.** [1] **Note:** 1 We controlled for offspring (i.e., age, sex, and birth order), parents (i.e., maternal and paternal education levels, mother's age and father's age, whether the child lived together with the mother, whether the child lived together with the father, father's employment status, and mother's employment status), and household (i.e., place of residence, household income in log scale, and number of children in the household) characteristics.

First, the associations between parental depression diagnosed during adolescence were consistently associated with cognitive outcomes at several points of measurement over 6 years and across a range of cognitive performance tests. Second, these associations were robust to adjustment of a wide range of potential confounders. Even after we added offspring retrospective cognitive measures during toddlerhood and offspring cognitive test scores measured in 2012 to the regressions, the associations remained robust. These results suggested that the observed associations between parental depression during adolescence and worse offspring cognitive performance might not be an extension of an association occurred in earlier lives, but more likely to have an independent pathway. Third, we observed that the association was modified by the offspring age of exposure in mathematics and vocabulary tests: As hypothesized, the association was stronger for younger adolescents (i.e., 10 to 11 years) in comparison to their older counterparts (i.e., 12 to 15 years), consistent with the theory and research on neuroplasticity [31]. Fourth, there was evidence to suggest a dose–response relationship, at least for maternal depression. Fifth, the finding that depression in either parent influenced cognitive outcomes is consistent with adolescent development theory which emphasizes the importance of both parents in shaping. Finally, there are plausible mechanisms to explain these associations [32,33]. For example, there is evidence that parental depression adversely affects parental ability to show affection, sensitivity, and responsiveness [2,34] and is associated with heightened risk of familial conflict [35], which are factors strongly related to adolescents' cognitive development and academic achievement [36].

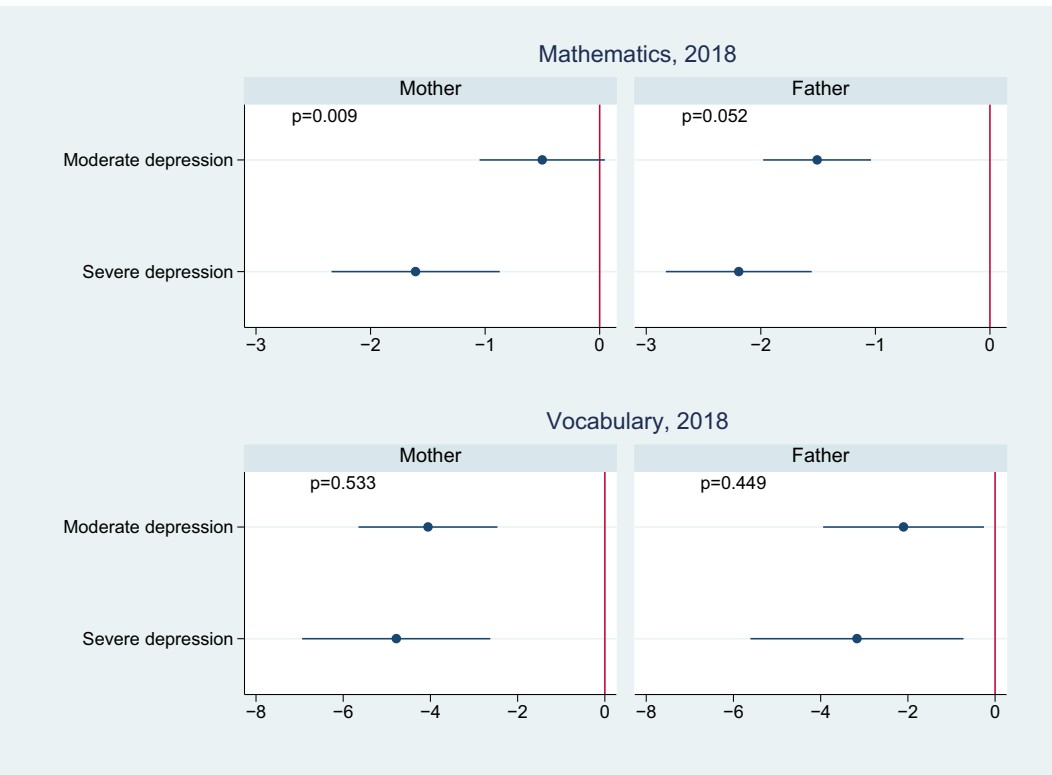

**Fig 7. Effect modification by severity of parental depression in 2012 on the association of parental depression in 2012 with offspring cognitive test scores in 2018.** [1] **Note:** 1 We controlled for offspring (i.e., age, sex, and birth order), parents (i.e., maternal and paternal education levels, mother's age and father's age, whether the child lived together with the mother, whether the child lived together with the father, father's employment status, and mother's employment status), and household (i.e., place of residence, household income in log scale, and number of children in the household) characteristics.

While there is insufficient existing evidence with which to compare our findings, we note that there is no consensus regarding how offspring sex modifies the association between parental depression and early child cognitive development. For instance, some studies observe that maternal depression has a significant adverse impact on boys' cognitive development in early childhood, but not on girls [5]. Other studies, by contrast, find the reverse pattern, marking girls as the more vulnerable group [10]. In our study, we observed that the cognitive performance of both boys and girls were equally associated with maternal depression, whereas the associations of paternal depression on boys and girls appeared more variable. The inconsistency of these results suggests the possibility that the differential results were chance findings or that there might be interactions with other factors specific to the exposure or outcome which need further exploration.

Several limitations of our study should be noted. First, we used self-rated depression symptoms as the exposure measure. However, this approach is the most commonly adopted one in epidemiologic studies and considered to be more scientifically sound than a binary diagnostic approach [17,18]. Second, we used retrospective cognitive measures based on parents' recall to control for offspring cognitive ability in earlier ages. These measures may suffer from recall bias. Thirdly, although the attrition rate of CFPS between 2 consecutive surveys was less than 15%, a total of 36% of the children identified in 2012 were incrementally lost over the 6 years of follow-up. However, there were no systematic differences between participants who successfully followed up with those who were lost, and sensitivity analyses using the multiple

**Table 4. The association of parental depression in 2012 with offspring cognitive test scores in subsequent years, adjusting for earlier cognitive ability along with offspring, parents, and household characteristics[1].**

| Depression status | Adjusting for cognitive ability in toddlerhood | | | | Adjusting for cognitive ability in 2012 | | | |
|---|---|---|---|---|---|---|---|---|
| | Mathematics | | | | Mathematics | | | |
| | 2014 (N = 918) | | 2018 (N = 517) | | 2014 (N = 1,230) | | 2018 (N = 714) | |
| | Coef (95% CI) | p-value | Coef (95% CI) | p-value | Coef (95% CI) | p-value | Coef (95% CI) | p-value |
| Maternal depression in 2012 | −1.220 (−1.554, −0.886) | <0.001 | −1.076 (−1.484, −0.669) | <0.001 | −0.581 (−0.834, −0.328) | <0.001 | −0.574 (−1.036, −0.111) | 0.031 |
| Paternal depression in 2012 | −1.765 (−2.143, −1.386) | <0.001 | −1.379 (−1.823, −0.936) | <0.001 | −0.892 (−1.210, −0.574) | <0.001 | −0.997 (−1.474, −0.519) | <0.001 |
| | Vocabulary | | | | Vocabulary | | | |
| | 2014 (N = 918) | | 2018 (N = 712) | | 2014 (N = 1,230) | | 2018 (N = 1,018) | |
| | Coef (95% CI) | p-value | Coef (95% CI) | p-value | Coef (95% CI) | p-value | Coef (95% CI) | p-value |
| Maternal depression in 2012 | −2.097 (−2.575, −1.619) | <0.001 | −1.365 (−2.446, −0.284) | 0.012 | −1.331 (−1.675, −0.987) | <0.001 | −2.607 (−3.696, −1.518) | <0.001 |
| Paternal depression in 2012 | −1.015 (−1.566, −0.464) | <0.001 | −3.386 (−4.723, −2.049) | <0.001 | −1.099 (−1.519, −0.679) | <0.001 | −5.297 (−6.644, −3.951) | <0.001 |
| | Immediate word recall, 2016 (N = 689) | | | | Immediate word recall, 2016 (N = 935) | | | |
| | Coef (95% CI) | | p-value | | Coef (95% CI) | | p-value | |
| Maternal depression in 2012 | −0.234 (−0.363, −0.105) | | <0.001 | | −0.279 (−0.387, −0.171) | | <0.001 | |
| Paternal depression in 2012 | −0.469 (−0.612, −0.325) | | <0.001 | | −0.465 (−0.598, −0.331) | | <0.001 | |
| | Delayed word recall, 2016 (N = 668) | | | | Delayed word recall, 2016 (N = 905) | | | |
| | Coef (95% CI) | | p-value | | Coef (95% CI) | | p-value | |
| Maternal depression in 2012 | −0.155 (−0.348, 0.038) | | 0.094 | | −0.454 (−0.612, −0.297) | | <0.001 | |
| Paternal depression in 2012 | −0.362 (−0.574, −0.150) | | <0.001 | | −0.263 (−0.464, −0.061) | | 0.011 | |
| | Number series test, 2016 (N = 694) | | | | Number series test, 2016 (N = 941) | | | |
| Maternal depression in 2012 | −0.539 (−0.873, −0.205) | | 0.002 | | −0.510 (−0.760, −0.260) | | <0.001 | |
| Paternal depression in 2012 | −2.558 (−2.918, −2.199) | | <0.001 | | −1.196 (−1.510, −0.881) | | <0.001 | |

Note

1 We controlled for offspring (i.e., age, sex, and birth order), parents (i.e., maternal and paternal education levels, mother's age and father's age, whether the offspring lived together with the mother, whether the offspring lived together with the father, father's employment status, and mother's employment status), and household (i.e., place of residence, household income in log scale, and number of offspring in the household) characteristics.

CI, confidence interval.

imputation approach showed consistent results. Fourth, we recognized that residual confounders, such as childhood depression, might have an impact on the association. Fifth, we cannot fully control for earlier parental depression on offspring cognitive development. Although we included offspring retrospective cognitive performance in toddlerhood and the cognitive performance in 2012 to control for the effects of earlier parental depression or deprivation on cognitive development, our estimates may be biased if earlier parental depression had an independent pathway to affect offspring cognitive performance in adolescence. Lastly, we cannot fully exclude comorbidity. It is possible that what we captured was actually comorbid depression among people with other mental or physical illnesses.

On the other hand, the current study has several strengths. First, this study has a longitudinal design, whereby we followed a population-based cohort for 6 years. Second, a wide range of concurrent and historic confounding variables is adjusted in our analysis, enhancing the

robustness of our results. Third, this is among the first studies to examine the association between parental depression and offspring cognitive development beyond the perinatal period. In comparison to a similar study by Shen and colleagues [10], which was in the context of Sweden, we were the first known study to be in a developing country. Moreover, we have adopted a variety of valid cognitive measures (e.g., vocabulary, mathematics, word recall, etc.) with each capturing a unique aspect of cognitive development. Our results strongly suggest that parental depression during adolescence is associated with adverse cognitive performance in later life. These findings greatly widen the developmental window of opportunity for interventions to promote cognitive development by preventing and treating parental depression across childhood and adolescence, with obvious implications for future health and well-being. More studies should be conducted in other settings to test the generalizability of our results.

## Supporting information

**S1 STROBE Checklist. STROBE Statement presenting checklist of items that should be included in reports of cohort studies.** STROBE, Strengthening the Reporting of Observational Studies in Epidemiology.
(DOC)

**S1 Table. Measurement of depression symptoms.**
(DOCX)

**S2 Table. Design of the cognitive tests.**
(DOCX)

**S3 Table. Summary table of the sample's characteristics at the baseline (2012).**
(DOCX)

**S4 Table.** Models on the association of parental depression in 2012 with offspring cognitive test scores in the following years: (a) Mathematics and vocabulary test scores; (b) Immediate word recall, delayed word recall, and number series test scores.
(DOCX)

**S5 Table.** The association of parental depression in 2012 with offspring cognitive test scores in the following years, effect modification by offspring sex: (a) Mathematics and vocabulary test scores in 2014 and 2018; (b) Immediate word-recall, delayed word recall, and number series test scores in 2016.
(DOCX)

**S6 Table.** The association of parental depression scores (continuous) in 2012 with offspring cognitive test scores in subsequent years: (a) Mathematics and vocabulary test scores in 2014 and 2018; (b) Immediate word recall, delayed word recall, and number series test scores in 2016.
(DOCX)

**S7 Table.** The association of parental depression in 2012 with offspring cognitive test scores in subsequent years, using multiple imputation to impute for observations loss of follow-up and observations with the values of key covariates missing: (a) Mathematics and vocabulary test scores in 2014 and 2018; (b) Immediate word recall, delayed word recall, and number series test scores in 2016.
(DOCX)

**S8 Table.** The association of parental depression in 2012 with offspring cognitive test scores in subsequent years, controlling for child CES-D score in 2012: (a) Mathematics and

vocabulary test scores in 2014 and 2018; (b) Immediate word recall, delayed word recall, and number series test scores in 2016. CES-D, Center for Epidemiologic Studies-Depression. (DOCX)

**S9 Table.** The association of parental depression in 2012 with offspring cognitive test scores in subsequent years, adjusting for parental cognitive scores measured in 2010: (a) Mathematics and vocabulary test scores in 2014 and 2018; (b) Immediate word recall, delayed word recall, and number series test scores in 2016.
(DOCX)

**S1 Text. Statistical analysis plan showing research question, data source and sample size, exposures, outcomes, covariates, statistical analyses, effect modification analyses, sensitivity analyses, and post hoc analyses in response to editors' and reviewers' comments.**
(DOCX)

## Author Contributions

**Conceptualization:** Zhihui Li, Wenjuan Qin.

**Formal analysis:** Zhihui Li, Wenjuan Qin.

**Investigation:** Zhihui Li.

**Methodology:** Zhihui Li, Wenjuan Qin, Vikram Patel.

**Supervision:** Vikram Patel.

**Validation:** Vikram Patel.

**Visualization:** Zhihui Li, Wenjuan Qin, Vikram Patel.

**Writing – original draft:** Zhihui Li, Wenjuan Qin.

**Writing – review & editing:** Zhihui Li, Wenjuan Qin, Vikram Patel.

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
