## [Editor Report · Decision Letter 0]

14 Jul 2020

Dear Dr Patel, 

Thank you for submitting your manuscript entitled "The effects of parental depression during adolescence on cognitive development in later life: a population based cohort study from China" for consideration by PLOS Medicine.

Your manuscript has now been evaluated by the PLOS Medicine editorial staff and I am writing to let you know that we would like to send your submission out for external peer review.

Kind regards,

Artur Arikainen,

Associated Editor

PLOS Medicine

---

## [Decision Letter · Decision Letter 1]

20 Aug 2020

Dear Dr. Patel,

Thank you very much for submitting your manuscript "The effects of parental depression during adolescence on cognitive development in later life: a population based cohort study from China" (PMEDICINE-D-20-03333R1) for consideration at PLOS Medicine. 

[LINK]

In light of these reviews, I am afraid that we will not be able to accept the manuscript for publication in the journal in its current form, but we would like to consider a revised version that addresses the reviewers' and editors' comments. Obviously we cannot make any decision about publication until we have seen the revised manuscript and your response, and we plan to seek re-review by one or more of the reviewers. 

We expect to receive your revised manuscript by Sep 10 2020 11:59PM. Please email us (plosmedicine@plos.org) if you have any questions or concerns.

We look forward to receiving your revised manuscript. 

Sincerely,

Artur Arikainen, 

Associate Editor 

PLOS Medicine

plosmedicine.org

1. Please address all of the reviewers’ comments below – further consideration of your manuscript will depend on satisfactory resolution of the concerns raised.

2. Title: Please update to: “Associations of parental depression during adolescence with cognitive development in later life, in China: a population based cohort study”

3. Abstract:

a. Please show age spread (eg. IQR) and % by sex.

b. Please list the main factors used for adjustment.

c. Please quantify the main results (with 95% CIs and p values).

d. In the last sentence of the Abstract Methods and Findings section, please describe the main limitation(s) of the study's methodology.

4. Please avoid causal language throughout your manuscript (‘effect’, ‘affected’, etc.), since your study cannot prove causation, eg. “Parental depression during adolescence had adverse effects on offspring cognitive development…” (line 62); instead refer to ‘association’, eg. “Parental depression during adolescence was associated with adverse offspring cognitive development…”.

5. In the Funding Disclosure, please list the funders, or state “The author(s) received no specific funding for this work”.

7. Please use the "Vancouver" style for reference formatting, and see our website for other reference guidelines: https://journals.plos.org/plosmedicine/s/submission-guidelines#loc-references. Citations should be in square brackets, and preceding punctuation.

8. Please correct “offsprings” to “offspring” throughout.

9. Please avoid referring to non-significant results as “trend(s)”, eg. line 285; simply state whether or not the result is significant.

10. Lines 364-368: Please remove the Funding and Competing Interests information, and instead include these in the online submission form.

11. Lines 370-373: Please move the Ethics information to the Methods section. 

12. Did your study have a prospective protocol or analysis plan? Please state this (either way) early in the Methods section.

13. Please ensure that the study is reported according to the STROBE guideline, and include the completed STROBE checklist as Supporting Information. 1 Please add the following statement, or similar, to the Methods: "This study is reported as per the Strengthening the Reporting of Observational Studies in Epidemiology (STROBE) guideline (S1 Checklist)." The STROBE guideline can be found here: http://www.equator-network.org/reporting-guidelines/strobe/ When completing the checklist, please use section and paragraph numbers, rather than page numbers.

------------

Comments from the reviewers:

Reviewer #1: I confine my remarks to statistical aspects of this paper. The general approach is fine, but I have two areas of concern to address before I can recommend publication.

First, the authors use causal language and this is not warranted. (Words such as "impact", "affect" and so on). 

Second, the authors should not categorize continuous independent variables such as parent age and CESD scores. In *Regression Moideling Strategies* Frank Harrell lists 11 problems with this and sums up "nothing could be more disastrous". I wrote a blog post illustrating some problems with this graphically. https://medium.com/@peterflom/what-happens-when-we-categorize-an-independent-variable-in-regression-77d4c5862b6c

Peter Flom

Reviewer #2: This manuscript reports the investigation of an important question about transgenerational effects of parental depression of offspring cognitive ability, and so the wider effects of parental depression on the life chances of the offspring. The sample is large and has many strengths, not least that it represents a population beyond the developed, Western world that contributes so much to the literature; the prospective design is appropriate and the analyses thoughtfully done. The manuscript is clear and the balance of reporting between the main text and supplementary material is good; the latter is necessary to support deeper reading.

For the editorial team the main question may be the extent to which this reports incremental advances by dint of the important sample, novel Chinese setting and methodologically elegant approach, or wehther it really breaks new ground. It is certainly a high quality paper. While the early developmental effects of parental (particularly maternal) depression on the child are well evidenced, later effects during puberty are less clear so this study is important. That said, the main paper illuninating this is pretty definitive, Shen et al., cited as #10, with a sample size of over a million and the ability to address the question left outstanding here as to whether the effects demonstrated have simply 'tracked' following a causally important effect earlier in life; there was also a well-cited commentary about the public health implications of their findings. This paper is cited appropriately and minor changes to the already excellent discussion to reflect what is definitive replication and what is advance could strengthen the work. Specifically, around line 308 I'd suggest the work contributes to preliminary evidence of causation, with the possibility having been around for some decades (eg the Murray, Cooper, Hay and other references).

For the authors, I have the following additional suggestions, again mainly to do with the discussion and interpretation, especially around the preliminaries of judging causality in terms of bias and confounding. I've mentioned tracking over development of cause and effect that may have occurred during early life, or may have operated causally only during a later 'window'. Shen et al addressed this to a degree, and the 'washing out' of the effect in this current paper during adolescence may fit with this.

The effect sizes are small: around one point. These were robust to adjustment for sensible confounders but the danger of residual confounding is ever-present in this kind of developmental observational epidemiology. One offspring characteristic not measured (that might be seen as a mediator) is childhood depression, either through a genetic effect, or a transgenerational response to shared family or wider environment.

On the small effect size, the fact that it appears more important for maths and vocab may be a function of the tests, themselves, given that these are much more granular (0-24 and 0-34) than for the other domains (0-10/15). I would make more of the consistent patterns and 'g' than the individual domains, but that's a contested area.

The authors mention attrition as a possible contribution to the changes in effects over the follow-up (age effects). One approach could be to model this, perhaps through imputation. Again, a contested area but a systematic approach to imputing exposure, outcome and/or confounders would shed light on the bias and might be reassuring. 

A small point: personally, I would be interested in the ethical pedigree of the data. The Harvard REC saw this as secondary analysis of anonymous data but what about the original collection? This is in no way meant to infer any problem; I am simply interested and believe it's a relevant aspect of describing the data in a gloabl health setting and encouraging openness.

PB Jones

Reviewer #3: This manuscript reports about an association between parental depression and children's cognitive development. There are several limitations.

Sampling.

The sampling is not clear. 16,000 households were sampled, but data were available for around 2000 households (?) in which depression data were available. It is important to know the difference between the panel study's sampling and available data. Were data constrained by parenting status, having children at home, refusal? More information is needed.

The study analyzes longitudinal data, with a retention rate of about 67%. I am aware of the great difficulty of recruiting and maintaining samples intact (see the important National Academy Science (US) report on the topic.). Still, one wonders if sound epidemiology can really be done on samples with so much attrition. At minimum, the authors need to provide details about whether there are systematic sources of attrition, and to evaluate the bias introduced. Although the authors claim that there were not many differences, the reason for this is most likely because there was so much attrition. That is, attrition effects actually become harder to detect when attrition is high, and that's because one loses people from both sides of the distribution, but the difference in the 'risk side' is obscured. It would be useful to see a visual representation of who was lost, as a function of their characteristics. 

Measurement.

The authors focus on depression, using the CES-D instrument. This is a widely used instrument. But, does it measure depression per se, or does it simply capture psychological distress more broadly? There is good reason to believe the latter. Moreover, simply because one administers a single mental health measure, it is not the case that the measure actually assesses that construct alone; in fact, the measure most likely picks up comorbid conditions. The reason this point is important is because the interpretive framework for this paper is depression, but there is way to establish that depression per, and not mental distress more generally, is being assessed. 

The authors note the scoring guidelines come from the American Psychological Association. That's not quite correct. The APA does have a link which describes the instrument, but that is not an APA guideline or endorsement; it is simply a description of the measure.

Covariates.

The most important variable that is necessary to aid interpretation of the results would be the cognitive status of the parents. This is, are we observing the putative effects of depression or are we observing the intergenerational transmission of cognitive ability (which happens to also be correlated with depression). Education is a fine covariate, but it is not an entirely adequate proxy for cognitive ability. 

Analysis. 

The effect modification analysis has yielded several interaction terms. Given the multiple test performed, these need to corrected for multiple testing. In addition, in performing these effect modification analyses, the authors need to estimate a fully saturated model, not simply one in which covariates are entered as main effects. There is actually a fairly sizeable literature on this topic, and not all of it says the same thing, but the fully saturated model should evaluated to see how it affects effect modification parameters.

Interpretation.

Given the above, especially the absence of a cognitive covariate from the parental generation and broader mental health assessment, it is not possible to interpret the results in terms of possible depression effects of cognitive development.

Editing.

A bit more attention to writing is needed, with occasional grammatical lapses. 

"Most research on the effects of parental depression on offspring cognitive outcomes have been limited…"

"the associations were temporal such that parental parental depression diagnosed during adolescence were…"

[LINK]

---

## [Decision Letter · Decision Letter 2]

23 Oct 2020

Dear Dr. Patel,

Thank you very much for re-submitting your manuscript "Associations of parental depression during adolescence with cognitive development in later life, in China: a population-based cohort study" (PMEDICINE-D-20-03333R2) for review by PLOS Medicine.

I have discussed the paper with my colleagues and the academic editor and it was also seen again by 3 reviewers. I am pleased to say that provided the remaining editorial and production issues are dealt with we are planning to accept the paper for publication in the journal.

[LINK]

We look forward to receiving the revised manuscript by Oct 30 2020 11:59PM. 

Sincerely,

Artur Arikainen, 

Associate Editor 

PLOS Medicine

plosmedicine.org

Requests from Editors:

1. Competing Interests: Please update the relevant sentence to our standard text: “VP is a member of PLOS Medicine's editorial board."

2. Please provide line numbers in the margin throughout.

3. Please give p values as “p<0.001” throughout, instead of “p<.001”. Please give exact values where greater than 0.001, including in Tables. In the Supplementary Information Tables, please do not report p=0.000; report as p<0.001.

4. The terms gender and sex are not interchangeable (as discussed in http://www.who.int/gender/whatisgender/en/ ); please use the appropriate term throughout.

5. Abstract:

a. Please quantify the main results (with 95% CIs and p values), including the following sentences:

i. “the associations between paternal depression and mathematical scores were also significant.”

ii. “These findings remained robust after adjusting for offspring earlier cognitive ability, offspring depression status, and parents’ cognitive ability.”

iii. “In line with the neuroplasticity theory, we observed stronger associations between maternal depression and mathematical / vocabulary scores among the younger adolescents (i.e., 10-11 years) than the older ones (i.e., 12-15 years).”

b. Conclusion: Begin with “In this study, we observed that…”

6. Introduction: We recommend you delete this sentence, or move it to the Discussion: “If our hypotheses proved to be correct, our findings would point to the importance of addressing parental depression in the context of cognitive development and educational attainment well beyond childhood.”

7. Throughout, please give ages as “aged x years”, rather than “aged x years old”.

8. Methods: Please mention that your study did not require additional specific approval from an ethics committee.

9. Results: Please refer to ‘association’ instead of ‘effect’ here and elsewhere: “For example, the effect of maternal depression on vocabulary scores in 2014 was -2.1 (95% CI: -2.6, -1.5) in those aged 10-11 years, comparing to -1.2 (95% CI: -1.6, -0.8) in those aged 12-15 years. Similarly, the effect of paternal depression on 2014 vocabulary scores was -1.7 (95% CI: -2.4, -1.0) in those aged 10-11 years, compared to -0.7 (95% CI: -1.1, -0.2) in those aged 12-15 years.”

10. Discussion: Please refer to ‘association’ instead of ‘effect’ here and elsewhere: “In our study, we observed that the cognitive performance of both boys and girls were equally associated with maternal depression, whereas the effects of paternal depression on boys and girls appeared more variable.”

11. Please remove spaces from within citation callouts throughout, eg “…hinder early childhood cognitive development [1,2].”

12. Please replace hyperlinks here with citations: “…guidance online regarding when an IRB application is required and the IRB Decision Tool.”

13. Results: Please quantify all comparisons with p values, as well as 95% CIs.

14. Tables: Where presenting data in Chinese Yuan, please also include an estimate in a more widely circulating currency, eg. USD.

15. Please upload the analysis plan and STROBE checklist as separate files, with the names “S1 Text” and “S1 Checklist”, respectively.

16. Please provide volume/issue/page or DOI for reference 35.

---

Comments from Reviewers:

Reviewer #1: The authors have addressed my concerns and I now recommend publication. 

Peter Flom

Reviewer #2: The authors have done a good job. Thank you for doing the additional analysis regarding attrition, for illuminating the ethics pedigree, and for the other changes.

Expunging the causal language is a great improvement, but 'The inter-generational impact of parental depression on cognitive development, established for perinatal maternal depression in early childhood, extends well into adolescence when the associations are visible for both parents.' is still pretty strong for an observational study (there will be residual confounding!). Perhaps 'Our results are consistent with an inter-generational etc.' or something a little less excited and in tune with the excellent discussion.

Reviewer #3: This is a responsive revision, for the most part. 

On the whole, I am underwhelmed by the research. I am not certain that it is especially novel. The one thing that makes it stand out is that it is non-Western. But, that's really it. 

A few outstanding items. 

The authors did not appear to understand my comment about fully-saturated models. Some statisticians argue that in order to interpret interaction effects, it is not enough to enter covariates, but one must enter ALL the interaction terms, including those with the covariates, into the models. The authors have only entered covariates as main effects, not as part of interaction terms. They should examine the implications of doing so. You might want to seek statistical advice on this score. 

The authors also did not understand my comment about comorbidity. Depression is not only comorbid with anxiety. It is comorbid with many conditions. For example, children who meet diagnostic criteria for conduct disorder are at heightened risk of depression. As such, I do not believe that one can actually speak of depression 'effects' in the absence of information about the many conditions which depression is correlated. This particular measure of depression, which taps into negative emotionality, is part and parcel of multiple psychiatric conditions.

[LINK]

---

## [Editor Report · Decision Letter 3]

26 Nov 2020

Dear Prof. Patel, 

On behalf of my colleagues and the academic editor, Dr. Peter B. Jones, I am delighted to inform you that your manuscript entitled "Associations of parental depression during adolescence with cognitive development in later life, in China: a population-based cohort study" (PMEDICINE-D-20-03333R3) has been accepted for publication in PLOS Medicine. 

PRODUCTION PROCESS

Before publication you will see the copyedited word document (within 5 business days) and a PDF proof shortly after that. The copyeditor will be in touch shortly before sending you the copyedited Word document. We will make some revisions at copyediting stage to conform to our general style, and for clarification. When you receive this version you should check and revise it very carefully, including figures, tables, references, and supporting information, because corrections at the next stage (proofs) will be strictly limited to (1) errors in author names or affiliations, (2) errors of scientific fact that would cause misunderstandings to readers, and (3) printer's (introduced) errors. Please return the copyedited file within 2 business days in order to ensure timely delivery of the PDF proof. 

If you are likely to be away when either this document or the proof is sent, please ensure we have contact information of a second person, as we will need you to respond quickly at each point. Given the disruptions resulting from the ongoing COVID-19 pandemic, there may be delays in the production process. We apologise in advance for any inconvenience caused and will do our best to minimize impact as far as possible.

EARLY VERSION

PRESS

PROFILE INFORMATION

Thank you again for submitting the manuscript to PLOS Medicine. We look forward to publishing it. 

Best wishes, 

Artur Arikainen, 

Associate Editor 

PLOS Medicine

plosmedicine.org